# Structural, Mechanical, and Decorative Properties of Sputtered TiN and Ti (N, C) Films for Orthodontic Applications; an In Vitro Study

**DOI:** 10.3390/ma14185175

**Published:** 2021-09-09

**Authors:** Victor Suciu, Armando Ferreira, Marcio A. Correa, Filipe Vaz, Daniel Munteanu

**Affiliations:** 1Department of Materials Science, Transilvania University, 500036 Brasov, Romania; victor.suciu@unitbv.ro; 2Centro de Física, Campus de Gualtar, Universidade do Minho, 4710-057 Braga, Portugal; armando.f@fisica.uminho.pt (A.F.); marciocorrea@fisica.ufrn.br (M.A.C.); fvaz@fisica.uminho.pt (F.V.); 3Departamento de Física, Universidade Federal do Rio Grande do Norte, Natal 59078-900, Brazil

**Keywords:** titanium nitride films, orthodontic applications, tribology

## Abstract

In this paper, we explore and modify the structural, mechanical, and decorative properties of films composed by TiN and Ti (N, C) with a wide range of N_2_ gas flow during the deposition in order to be used on orthodontic systems. The films were grown using reactive DC magnetron sputtering from a pure Ti target and customized with C pellets onto Si and stainless steel 316L substrates. The structural properties were studied using X-ray diffraction and scanning electron microscopy, while the mechanical ones were obtained through hardness, elastic modulus, and friction coefficient. Moreover, the wear rate has been measured under an artificial saliva medium to simulate the oral cavity. The color of the films deposited onto stainless steel 316 L substrate was characterized through CIELab color code. Our findings show that the addition of N_2_ and C in the Ti matrix improves the mechanical properties of the films. With the increase in the amount of N_2_ and C, the hardness reaches a value of 739 HV, higher than the one reported in the literature (600 HV), a low value of the coefficient of elasticity (8.0 GPa), and also a low friction coefficient (0.30). Moreover, with the addition of N_2_ and C in the Ti films, the color of the films changes from metallic aspect until “with” gold, which means that our coatings exhibit versatile mechanical and color characteristics to be used in orthodontic wires applications.

## 1. Introduction

Orthodontics is centered on correcting and preventing dental-facial anomalies such as maligned teeth or improper positions of the jaws and, consequently, incorrect occlusion. In the last decades, orthodontic appliances have known a significant development aided by the metal manufacturing technologies that have become available [1,2]. These appliances consisted of attachments that are bound to the surface of the tooth, as shown in Figure 1; they are called brackets and wires produced from nickel (Ni) and titanium (Ti), the “nitinol”, and become one of the most used materials for orthodontic wires, along with stainless steel and β-Ti [3].

Moreover, the entire principle of orthodontics is based on the movement of the teeth. To initiate this, force must be applied using the appropriate application point, direction, and magnitude.

The force is provided either by wires that are engaged in the bracket or by auxiliaries, such as elastics or springs. Shortly after the application of force tipping, the movement of the tooth starts, which will create an angle between the bracket slot and wire. When this contact reaches a specific contact point, adhesion between the metallic surfaces will occur, resulting in friction resistance to sliding [4,5].

Consequently, different works have shown a high interest for the subject and involved coating procedures for orthodontic appliances in order to reduce the friction upon contact points [6,7,8], to improve the adhesion [3,9], and to modify/control the corrosion of the metal in the oral cavity [10,11,12]. For instance, the use of ZnO nanoparticles was successfully employed to reduce the friction between brackets and orthodontic wires [13]. The nanoparticles have no cytotoxic effect and also can be considered antibacterial, but as the author specified, new studies are required [13]. In this study, the authors reached a friction coefficient of around 0.2 for the ZnO with nanophase [13].

To improve the adhesion properties, stainless steel wires were coated with inorganic fullerene-like nanoparticles of WS2 [9]. The wires coated with these nanoparticles revealed a substantial reduction in friction during tooth movement [9]. Another work shows the efficiency of titanium aluminum nitride (TiAlN) and tungsten carbide/carbon (WC/C)-coated wires to determine the friction and load-deflection rate compared with uncoated β-Ti wires [14]. The results show that the WC/C-coated wires have better properties than TiAlN, but both coated variants present better results in terms of friction resistance and load deflection than the uncoated ones. There are also many other research works, which provide results regarding the corrosion resistance of orthodontic devices. For instance, coating the wires with protective films such as TiAlN, N doped TiO_2_, or simply epoxy coating or also titanium nitride (TiN) thin films to improve the corrosion resistance of orthodontic wires [15,16,17,18].

Another approach not yet explored is the color of the arch wire. Thin films based on TiN can create light colors, which enable the patient the possibility of choosing the color of the arch wire, which may be appealing to the patients, thus increasing the acceptance of metal-type orthodontic appliances. TiN thin films were well known and used in industries, which imply cutting procedures due to their high level of hardness, but also for its decorative properties, exhibiting a wide range of colors that can be an advantage because of the different light reflectivity properties near the teeth enamel in comparison with uncoated metal orthodontic appliances [19,20,21]. Based on our former expertise and consulting other different research group results from TiN could have various colors going from metallic gray to gold or even brownish red when the N_2_ flow is raised. At the same time, relating to TiCN coatings, depending on the percentages of nitrogen and carbon, the color could be gray or gray-blue (for those with higher carbon percentages) or brownish (for a high amount of nitrogen).

In this sense, for this kind of application, it is interesting to investigate materials (compounds) in which we can improve not just the friction, adhesion, and resistance to the oxidation on the oral cavity but we can also insert new functionalities in the wires, such as decorating them with a specific color.

Motivated by this issue, we verify a real challenge for the dental domain to use thin films for orthodontic applications. Besides the possibility of modifying the wire colors (the decorative aspect), there is also the possibility to reduce friction between tooth enamel and material and also to increase the wires’ corrosion resistance. There are not many studies in the technical literature that correlate all these aspects for certain films.

Within the framework of this paper, we conduct a systematic study of these aspects, which brings forth. Here we explore two sets of samples, TiN and Ti (N, C) films grown onto stainless steel 316 L and (100)-oriented Si substrate by using reactive magnetron sputtering.

For this purpose, systematic studies of structural and mechanical properties of these systems in a simulated oral cavity have been performed. Our findings bring an interesting way to explore/modify the hardness, elastic modulus, and friction coefficient and, at the same time, introduce decorative aspects in the orthodontic systems by modifying the color of the films.

## 2. Materials and Methods

### 2.1. Sample Deposition

For this study, the TiN and Ti (N, C) films were deposited by DC reactive magnetron sputtering on stainless steel 316 L and (100)-oriented Si substrates. The depositions were carried out by using the following parameters: base pressure of 5 × 10^−4^ Pa, work pressure of 4 × 10^−4^ Pa, Argon gas flow of 25 sccm, time deposition of 600 s. The DC source was set at 400 V and 1.5 A. For all deposition processes, the substrates were kept almost at room temperature. By using these parameters, we grow two sets of samples. The first one is TiN films, in which we vary the N_2_ flow of 1.0, 2.4, and 7.0 sccm. The second set of samples are Ti (N, C) films with N_2_ flow of 2.4, 2.5, and 3.0 sccm. For this set, the Ti target was customized with carbon pallets with resulting in a coverage area of 66 mm^2^. To verify the dependence of structural and mechanical properties with the carbon pallets concentration, we produced Ti (N, C) films with a N_2_ gas flow of 2.4 sccm and decorated the Ti target with a covered area of 132 mm^2^. Using these procedures, we produced the samples presented in Table 1.

### 2.2. Structural Characterization

X-ray diffraction (XRD) measurements were performed to obtain the phase composition of the films. The results were acquired from the samples deposited on Si substrate, using a diffractometer Philips PW 1710 (Bruker, Billerica, Massachusetts, EUA) equipped with a CuKα radiation source, operating in a Bragg–Brentano configuration. The measurements were obtained using a 2θ range of 20° up to 90° with a step of 0.04°, with a voltage of 40 kV and a current of 40 mA.

Additionally, based on the XRD spectra, the average crystalline grain size (D) was calculated using Scherrer’s equation [22,23],
(1)$$D=\frac{k\lambda }{d\left(\cos\theta \right)}$$
where λ = 1.5406 Å, k is the crystalline shape factor, d is the full width at half maximum intensity (FWHM), and θ is the Bragg angle.

Cross-sectional and top-view scanning electron microscopy (SEM) images of the films were obtained from an EDAX-Pegasus X4M (Ametek Inc., Berwyn, PA, USA) scanning electron microscope. The average thickness, as well as the observation of the columnar feature, was observed by analysis in cross-section micrographs.

### 2.3. Physical Characterization

To evaluate the mechanical properties (hardness—H, elastic modulus—E, and adhesion critical loads—Lc2, Lc3), the deposited films were subjected to nanoindentation and microscratch tests in ambient atmosphere, using a nanoindentation tester (NHT2 nanoindenter (CSM Instruments/Anton Paar, Peseux, CH-2034, Switzerland) equipped with a three-sided diamond pyramidal Berkovich indenter tip with an average curvature of about 100 nm) and a microscratch tester (MST, equipped with a diamond Rockwell-type indenter with a tip radius of about 100 μm–(CSM Instruments/Anton Paar, Peseux, CH-2034, Switzerland). The indenter loading process was linear at a rate of 0.03 N/min, with a maximum load of 1.5 mN. The acquisition rate was 10 Hz, and the nanoindenter speed for approaching and retracting was 2000 nm/min. For statistical relevance, each data point was determined from the average of at least thirty measurements. The error calculation is obtained from the standard deviation measurements. The nanoindentation data were analyzed with the Oliver–Pharr model [24]. From a general point of view, the error bar is small, with few exceptions. Thus, to turn clear the figures, just the means values are presented.

The microscratch tests supposed a progressive load, with a start force of 0.03 N, a maximum load of 20 N, and a loading load rate of 10 N/min. The horizontal measurements length was 3 mm, and the sliding-horizontal speed was 1.5 mm/min. Here, each sample was subjected to at least three scratch tests, and the values for critical loads were averaged. The critical load values were obtained base on an optical analysis of the scratch tracks. According to the theoretical support of the scratch method, Lc2 represents the load corresponding to the first delamination of the film, and Lc3 is the load responsible for the delamination of more than 50% of the film from the scratch track.

Taking into consideration the orthodontic application of these thin films, meaning that the environment of use is the human oral cavity, the wear behavior has been evaluated in special wet conditions, using artificial saliva with the following ingredients: purified water, glycerin, 1% L-arginine, 0.2% sodium hyaluronate, polyvinylpyrrolidone, calcium chloride, and potassium.

The wear behavior of the deposited films was evaluated using a ball-on-disk tribometer (CSM Instruments/Anton Paar, Peseux, CH-2034, Switzerland), in rotation mode, against 100 Cr6 (AISI/ASTM 52,100) 6 mm diameter steel balls; the wear testing protocol was: linear speed 8 cm/s, normal load 1 N, stop conditions 24 m.

Both the samples and the steel balls were cleaned with isopropanol before each test in order to remove the surface impurities. The dynamic friction coefficient values were acquired using an LVDT transducer (linear variable differential transformer).

To remove any surface contaminants, each wear track has been cleaned, first in ethanol, second in an ultrasonic bath, and last using compressed air. A Surtronic 25 profilometer (Taylor-Hobson, Leicester, LE4 9JQ, UK) was used to evaluate the wear track section profile in six different positions; base on these results, the average wear rate K was calculated, with the following equation:(2)$$K=\frac{V}{\mathrm{Fl}}$$
where V is the volume of material removed from the sample in mm^3^, F is applied load (N), and l is the sliding length (m).

The colors of the films were characterized according to the International Commission on Illumination CIELAB color system. For this analysis, a commercial colorimeter CM-2600d Minolta (Konica Minolta, Marconibaan, MR Nieuwegein, The Netherlands) working with a wavelength range of 400–700 nm and a diffused illumination (D65 light source) was used. The results are represented according to the CIELAB color space scale. For all the samples, three readings were performed at different locations on the surface, at an 8° viewing angle, and the observer places at a 10° angle. The device was equipped with a 52 nm diameter integrating sphere and 3 pulsed xenon lamps.

## 3. Results and Discussion

### 3.1. Structural Properties

Figure 2a shows the structural results of the TiN set of films. The figure shows an evolution of the structural behavior as the N_2_ gas flow increases. For the TiN film deposited with a low N_2_ gas flow of 1 sccm (A1 film), it is possible to observe an intense peak located at 2θ ≈ 38.3°, associated with the well-defined (200) preferential direction of hcp α-Ti structure (space group P63/mmc). Despite the low N_2_ gas flow, it is possible to observe a weak peak at 2θ ≈ 36.7° related to the (111) diffraction pattern of TiN (ICSD card no. 64,906). The low N_2_ gas flow in the reactive sputtering generates defects and interstitial sites resulting in compressive stress, which is reflected in a displacement to the left of the diffraction peak in the measurement (see the dashed red line in Figure 2a). For the A2 film, in which the N_2_ gas flow is 2.4 sccm, we verify the fcc TiN phase, with a preferential growth on (220) direction due to the intensity peak observed at 2θ ≈ 62.12° (ICSD card no. 64,906). Moreover, it is possible to verify the well-defined peaks at 36.86° and 42.82° related to (111) and (200) TiN diffraction patterns, respectively. As can be observed by the blue dashed vertical lines, Figure 2a, the experimental TiN peaks present a slight shift, indicating vacancies/defects in the films resulting in an internal uniform compressive stress on the film due to the modification of the lattice parameters. Increasing further the N_2_ gas flow to 7 sccm (A3 film), we observe a decrease in the peak’s intensity, leading to the (200) TiN peak suppression. For this N_2_ gas flow, the (220) diffraction pattern intensity decreases, while the (111) one increases, showing rather undefined preferential growth direction in this case. These mechanics can be associate with the high N_2_ gas flow, in this case, leading to the saturation of N_2_ on the film, limiting the formation of a preferential direction. From a general perspective, these results allow us to modify the film structures by changing the N_2_ gas flow, which will modify the mechanical properties as discussed below. Our structural findings are in complete accordance with the one observed in the literature, in which the Ti and TiN films are studied [25,26].

Figure 2b depicts the structural results obtained for the Ti (N, C) films. For the B1, B2, and B3 films, in which the N_2_ gas flow is 2.4, 2.5, and 3.0 sccm, the structural results seem to present similar results to that one observed for the A2 and A3 films. However, the modification on the preferential growth direction can be verified. While the B1 and B2 films show fcc TiN (111) preferential growth direction, the B3 sample shows a considerable increase in the (220) TiN direction. Moreover, the TiN diffraction patterns peaks present a shift to the left in relation to the ICSD card no. 64,906 peak position. These features can be associated with two distinct mechanisms: (i) Interstices in the TiN structures due to the insertion of C atoms in the system. Consequently, the modification in the lattice parameter results in tensile stress. (ii) The formation of the TiC phase in the system, resulting in an overlap of the TiN and TiC peaks in the XRD patterns. This last mechanism is more evident for the B4 film, where the C pallet is covering an area of 132 mm^2^. Although the B4 film presents (111) TiN preferential growth direction, it is possible to verify a considerable decrease and enlargement of the peaks. As depicted by the green dashed line, this feature should be associated with the TiC phase formation (ICSD no. 44,494), leading to an overlap between the TiN and TiC peaks. Moreover, the reduction in the peaks intensity of the diffractions patterns with the increase in the C pallets area could be a result of the development of amorphous carbon agglomerate around the TiN grains.

From the XRD analysis, we can verify the average grain size of the studied films. In some nanocrystalline materials, extrinsic factors, but also intrinsic factors such as grain size, can affect friction and wear, as smaller grain sizes can decrease the friction coefficient [27]. Here, the grain size for each sample is obtained using Scherrer’s equation [23]. For the calculation, we consider λ = 1.54 Å and k = 0.9. In general, the TiN films present a grain size of around 9 Å, while for the Ti (N, C), values of approximately 17 Å are observed for the samples in which the C pallet area is 66 mm^2^, decreasing to around 10 Å for the B4 films, where the C pallets area is 132 mm^2^. This behavior can be associated with a core-shell-like behavior, in which the C conglomerate evolves the TiN grain, limiting the increase.

Representative results of the surface morphology, microstructure, and thickness were evaluated by SEM micrographs using cross-section and top view as represented in Figure 3. The results show an average thickness of 1.1 μm for the TiN set of samples and around 0.7 μm for the Ti (N, C) set of samples. From these results, the well-defined vertical columnar growth of the films is clear for both sets of films. According to Mahieu’s proposed model of the extended structure zone [28], sample A3 can be classified as a Zone Ic. These kinds of microstructures, fitting the Ic zone of the Mahieu’s ESZM, are often associated with crystalized nanostructures that, for both systems, coincide with the formation of the Ti intermetallic phases. This type of zone can be noticed to have nucleation and formation of crystallite islands, clear faceted grains that have no diffusion between them, and columns separated by grain boundaries. The Ti (N, C) samples (B2 and B4) show a more predominant T zone, with V-shaped and faceted columns. The mobility of the adatoms is high enough to allow diffusion between grains. This can be compared to zone II in the Thornton model, with a crystallographic out-of-plane orientation corresponding to the fastest growing direction [29].

### 3.2. Mechanical Characterization

Usually, in studies based on elastic modulus for orthodontic wires, the values founded range between 150 and 229 GPa [30,31]. In order to evaluate the hardness and elastic modulus in our samples, nanoindentation measurements were made to the TiN and Ti (N, C) films. The results are presented in Table 2; starting from them, the elastic modulus values increase from 112 GPa (for sample A1) and reach a maximum value of 159 GPa (for sample B2). The best samples to have a suitable matching between the coating and the substrate are films A3, B2, B3, and B4. The sample that has the highest hardness value, at 8.0 GPa, is the B4 film, which is sputtered at 2.5 sccm N_2_ gas flow and with 132 cm^2^ of carbon pellets, having the highest microhardness from the studied systems. This result is in accordance with the structural ones, in which the C agglomerates seem to generate a core-shell-like structure, limiting the growth of the TiN grains.

In dental literature, the results of Vickers hardness (HV) are in agreement with our findings, where several reports demonstrate the highest hardness around 484 up to 600 HV [32]. The clinical implication of hardness data is associated with the arch wire itself and the matching with mechanical properties of the bracket. Since hardness is an indication of the material resistance in plastic deformation, the higher the hardness of the alloy, the higher the resistance to plastic deformation. In this study, the hardness of the sample B4 has shown much higher hardness, 739 HV, than the values (600 HV) reported in the literature [32]. On the other hand, the elasticity of the TiN and Ti (C, N) samples have a maximum value of 8.0 GPa, and the literature reveals a higher value of 64 GPa [32]. This means that our coatings exhibit versatile mechanical characteristics to be used in orthodontic wires applications.

### 3.3. Friction Coefficient and Wear Rate

The experimental results show that the friction coefficient (µ) is not affected by the hardness of the substrate. For a given film, identical test results were obtained for the two substrates. For instance, in the TiN (A2 sample) and Ti (N, C) (B1 sample) films, the starting values of the friction coefficient are 0.28 and 0.47, respectively. The titanium nitride film coefficient increases gradually during sliding until it reaches a plateau, while Ti (N, C) films hit a plateau from the very start; 0.296 for TiN and 0.265 for Ti (N, C). The variation of the friction coefficient is relatively stable for the sample A2 and B1, regardless of the distance, while the sample B1 exhibits a significantly lower static friction coefficient based on the starting point. For Sample the A2, the friction coefficient is high at the start of the test, approximately 0.47, and decreases gradually to a low average value of 0.296 after run-in. By comparison, sample B1 does not exhibit a run-in period; the coefficient is low at the initial stage of sliding and remains at this level for the entire sliding distance. These representative results describe the behavior for the two sets of films studied here.

The friction coefficients (μ) as a function of N_2_ gas flow can be observed in Figure 4. In Figure 4a, we depict the results observed for the TiN set of films, in which we observe a decrease in μ as the N_2_ gas flow decreases from 1 down to 2.4 sccm, followed by a stabilization for the N_2_ gas flow of 7 sccm. Similar behavior can be observed for the Ti (N, C) films, as shown in Figure 4b. The open square depicts the results obtained for the Ti (N, C) films deposited at 2.5 sccm with a covered area of 132 mm^2^ of C pallets.

The insertion of C pallets with 66 mm^2^ leads to a decrease in the μ, irrespectively of the N_2_ gas flow employed. Considering N_2_ gas flow from 2.4 sccm, it is possible to verify a stabilization of the μ.

Considering the Ti (N, C) film with 132 mm^2^ of C pallets area, an increase in the μ to 0.30 can be observed in our results. The insertion of the C pallets allows us to decrease the friction coefficient, although the hardness measurement shows a weak modification in their properties when compared to TiN films.

Moreover, considering the present results, we verify relatively low wear rates, as can be seen in Figure 5. The values for the TiN set of films are presented in Figure 5a. The results present a slight increase from $7.8\times 10^{-6}{\;\mathrm{mm}}^{3}/\mathrm{Nm}$ up to around $98\times 10^{-6}{\;\mathrm{mm}}^{3}/\mathrm{Nm}$ for the films A1 and A3, respectively. On the other hand, the wear rate observed for the Ti (N, C) set of samples, with 66 mm^2^ of C pallets decrease to around $5.6\times 10^{-6}{\mathrm{mm}}^{3}/\mathrm{Nm}$ for the B1 sample, reaching $6.1\times 10^{-6}{\mathrm{mm}}^{3}/\mathrm{Nm}$ for the B2 samples.

The modification of the C pallets area to 132 mm^2^ (B4 film) allows us to obtain values of $6.8\times 10^{-6}{\mathrm{mm}}^{3}/\mathrm{Nm}$ for the wear rate. It is important to point out that all these wear characterizations were obtained in a simulated oral cavity, as discussed before.

Moreover, and taking into account the results obtained for coatings of commercially available aligning (NiTi) aesthetic arch wires from four different companies [33], the results presented show that the least mean friction coefficient value is 0.672 and the maximum value is 1.104 [33]. These results indicate that the commercially available arch wires have a friction coefficient significantly higher compared with our results. This means that our findings concerning mechanical characterization bring to light an exciting way to tuning the mechanical properties to given applications on orthodontic systems.

### 3.4. Color Coordinates

The use of titanium and nitrogen is related to decorative thin films. Thus, we go beyond the structural and mechanical characterization of the studied samples. Our films turn out to be able to modify the color of the arch wires of the orthodontic systems. The modification in the colors is verified by using the color coordinates of the CIELab color space system as a function. This property was measured as a function of N_2_ gas flow.

The numerical results are presented in Figure 6a,b. All samples have a moderate L* value, which can mean that the films have successfully blocked out the metallic L* value, which is known to have high values of brightness (L*), which are determined by interactions between incident photons and free electrons. Regarding chromaticity, the coordinates between the two types of samples are relatively close to each other regardless of the function of nitrogen gas flow. When the gas flow increases from 1 to 7 sccm, the values of a* and b* show no significant changes.

The brightness of the samples decreased significantly by increasing the N_2_ gas flow between 1 and 2.3 sccm. The L* value is found above 40 while increasing the flow determined the L* value drop approximately to 30 and plateau at this level regardless of the increased gas flow. Accordingly, to the brightness L* coordinate for teeth enamel, the best chromatic results from metallic appliances are obtained if the L* value is at the same level or below this value in order not to reflect the incident light brighter than the teeth itself. Figure 6c shows representative spheres in which it is possible to verify the color in a real system. For this purpose, the CIELab color codes were converted into hexadecimal ones, and the results are simulated. These remarkable results enable the patient to choose the wires and brackets color without modification on the mechanical properties of the system.

The samples with the lowest value of carbon present the best colorimetric characteristics for blending in with the enamel chromatic levels, exhibiting champagne color with the aesthetic of “white” gold. This behavior was already observed in a different set of samples produced by the authors using other deposition methods such as magnetron sputtering-arc evaporation process or cathodic arc [34,35]. It is reported that only human saliva can be used to test the magnitude exactly or to rank the efficiency in simulating the orthodontic sliding [36]. To assess friction and its coefficients in the wet environment, human saliva is most appropriate. For this present study, a wet solution containing artificial saliva was used. It has been demonstrated the favorable characteristics of titanium nitride and titanium carbon nitride films with regard to corrosion resistance, wear resistance, and increased hardness. This study showed that also the colorimetric and frictional properties could be an advantage for orthodontic applications where low friction coefficients can enhance the teeth movement, thus shortening the treatment times. These results suggest that the friction coefficient of the orthodontic wires was improved by TiN or Ti (N, C) films. Sliding mechanics is widely used in orthodontics treatment. The friction at the surface between bracket and wire may interrupt or significantly slow down the movement. The low level of friction may increase tooth movement efficiently. Therefore, low friction wires are desired. The TiN films exhibit low friction, suggesting that TiN-coated wire could be useful in orthodontics treatment.

## 4. Conclusions

This study has demonstrated that TiN and Ti (N, C) films produced by DC reactive magnetron sputtering can improve the structural and mechanical properties of the films for future orthodontic devices. Furthermore, the mechanical tests exhibit adequate mechanical characteristics, considering the previously mentioned application. These films had similar thicknesses in the range of 700–1100 nm. Overall, the Ti (N, C) film had the best surface properties among the tested samples, including a significantly lower and more stable coefficient of friction. Compared to the usually employed materials, the TiN film provided a higher frictional force but better mechanical properties. Moreover, we explored the TiN and Ti (N, C) films as decorative properties for orthodontic applications. For this purpose, we modified the color of the films and measured the properties by using CIELab color characterization. Our findings bring an exciting way to explore/modify the hardness, elastic modulus, and friction coefficient and, at the same time, introduce decorative aspects in the orthodontic systems by modifying the color of the films.

## Figures and Tables

**Figure 1 materials-14-05175-f001:**
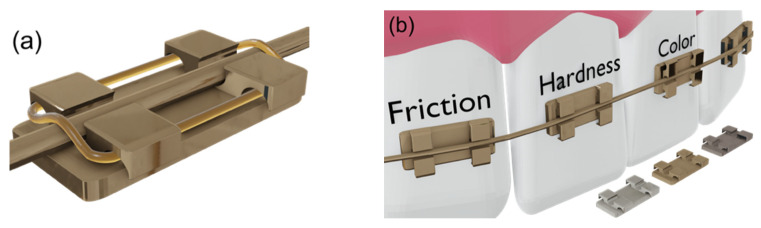
(**a**) Orthodontic brackets, wires, and rubber bands representation. (**b**) The main parameters studied in these systems.

**Figure 2 materials-14-05175-f002:**
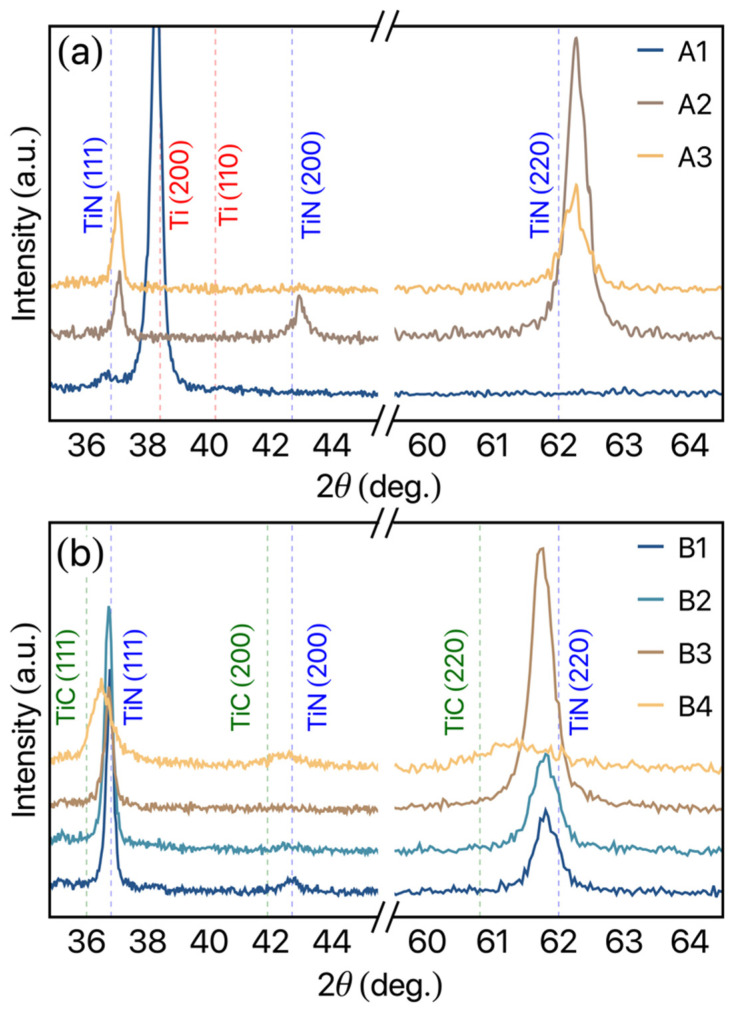
X-ray diffraction pattern for the (**a**) TiN films and (**b**) Ti (N, C) films. The peaks were indexed by using the ICSD cards no. 253,841, 64,906 and 44,494 for the Ti, TiN, and TiC phases, respectively.

**Figure 3 materials-14-05175-f003:**
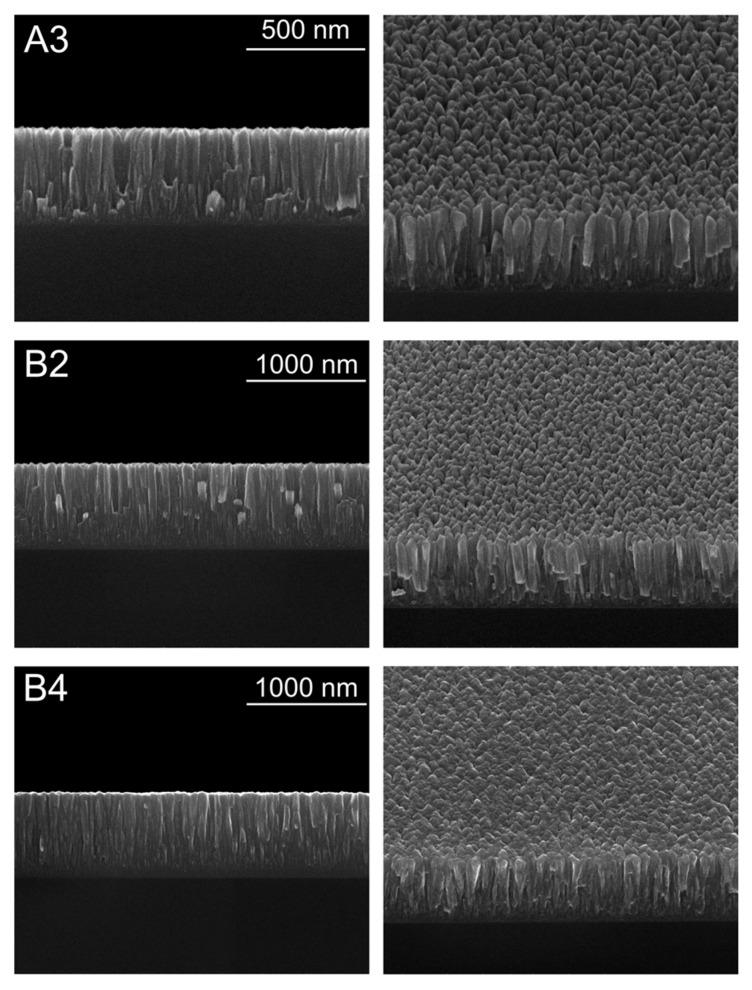
Representative SEM images for the set of studied samples. In particular, the left column shows the cross-section and the right column the top view for the A3, B2, and B4 studies films. The scale bar indicated for the cross-section images is the same for the top-view ones.

**Figure 4 materials-14-05175-f004:**
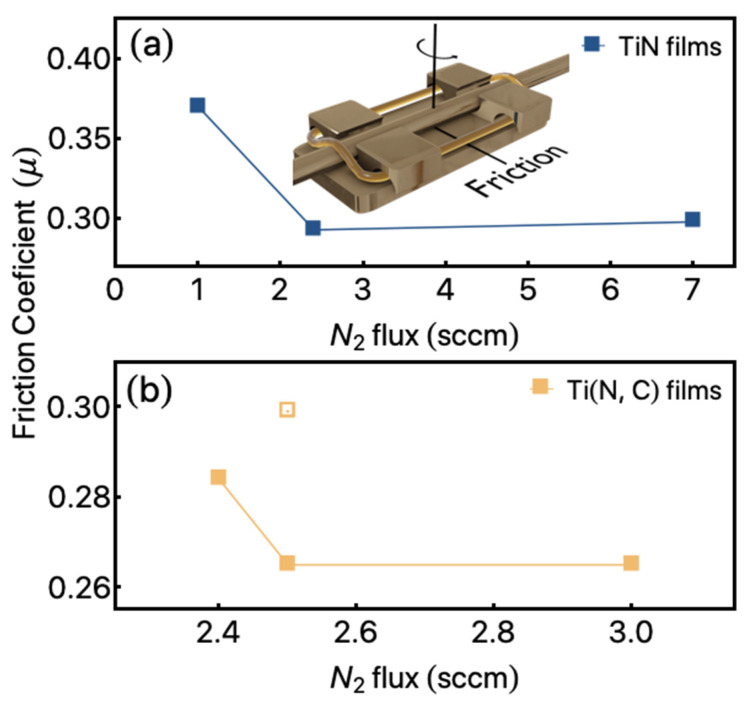
Friction coefficient as a function of N_2_ gas flow for the studied films. (**a**) Results were observed for the TiN films. The inset shows a characteristic bracket and the respective wire indicating the friction in the real system. (**b**) Results were obtained for Ti (N, C) films. Especially the substrate tested in identical conditions presented µ = 0.91. The open symbol shows the results observed for the Ti (N, C) film in which 132 mm^2^ of C pallets were used.

**Figure 5 materials-14-05175-f005:**
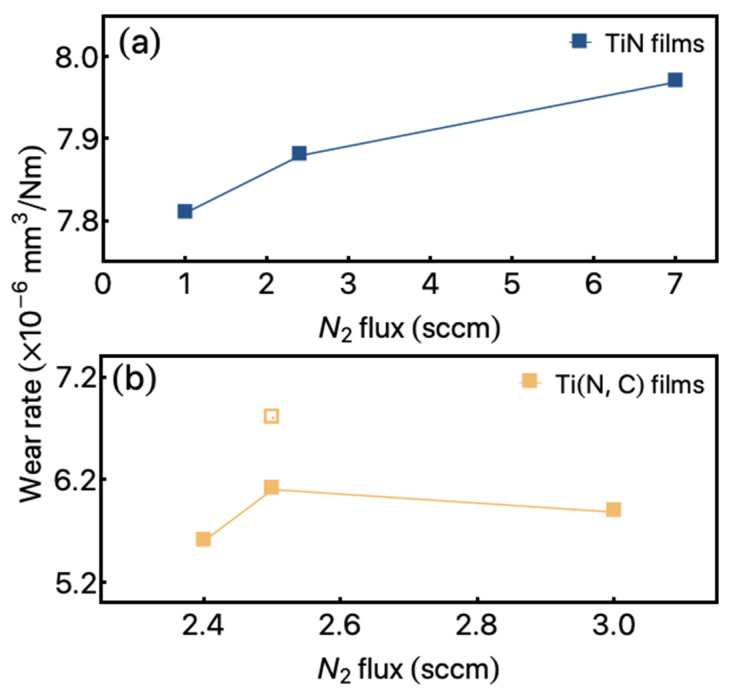
Wear rate as a function of N_2_ gas flow for the studied films. (**a**) Results obtained from TiN films. (**b**) Results observed for the Ti (N, C) films with 66 mm^2^ (close squares) and 132 mm^2^ (open square) of carbon pallets.

**Figure 6 materials-14-05175-f006:**
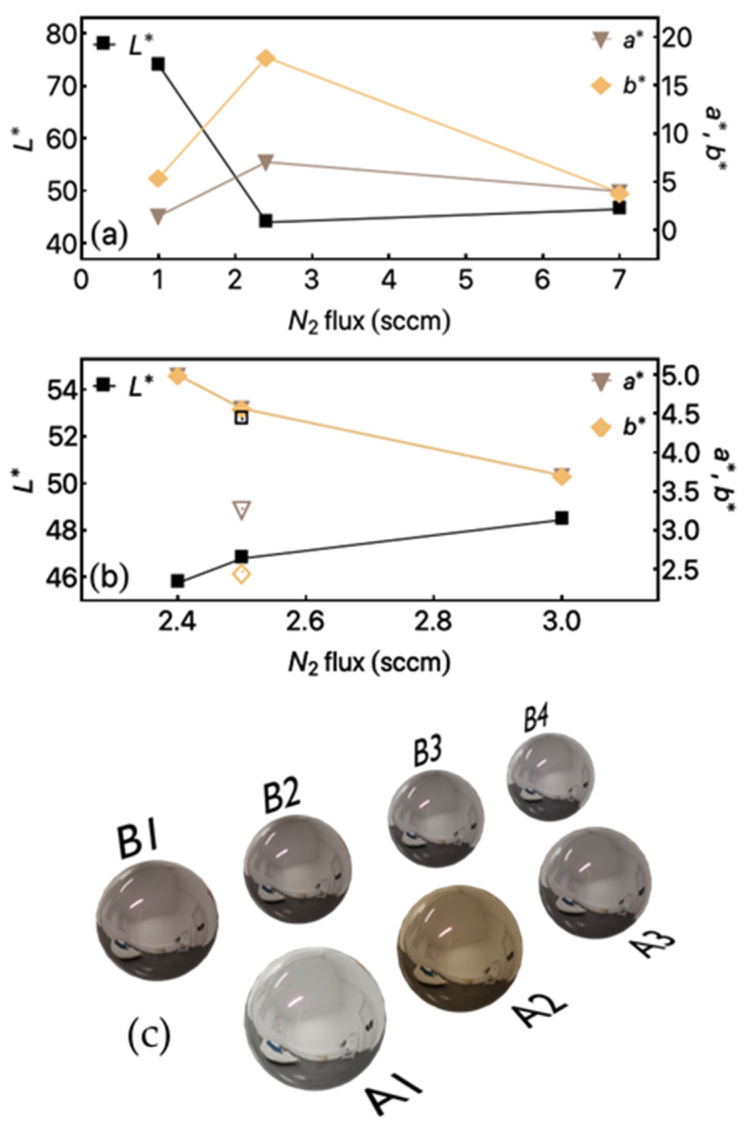
L*, a*, and b* factors for the studied films as a function of N_2_ gas flow (**a**) TiN films (samples A1, A2, and A3). (**b**) Ti (N, C) films. The closed symbols depict the films in which 66 mm^2^ of C pallets were used (samples B1, B2, and B3). The opened symbols depict the results obtained for Ti (N, C) films in which were used 132 mm^2^ of C pallets (Sample B4). (**c**) Representation of the CIELab colors in a metallic system. For this representation, the CIELab color codes were converted in the hexadecimal code, and the results were simulated on the representative spheres.

**Table 1 materials-14-05175-t001:** Parameters of DC reactive sputtering deposition process of TiN and Ti (N, C) films. The N_2_ gas flow can be converted to SLPM, taking the conversion factor of ×10^−3^. For instance, for the A1 films, the N_2_ gas flow is 1.0 × 10^−3^ SPLM.

Dep. Parameter	TiN			Ti (N, C)			
	A1	A2	A3	B1	B2	B3	B4
N_2_ gas flow (sccm)	1.0	2.4	7.0	2.4	2.5	3.0	2.5
C pellets’ area (mm^2^)	--	--	--	66	66	66	132

**Table 2 materials-14-05175-t002:** Mechanical characteristics of coated samples. H—indentation hardness; HV—conversion to kgf/mm^2^ from GPa; E—indentation elastic modulus; Lc2—second critical load; Lc3—third critical load.

Parameter	TiN			Ti (N, C)			
	A1	A2	A3	B1	B2	B3	B4
H (GPa)	4.4 ± 1.3	3.0 ± 0.5	5.2 ± 0.7	4.3 ± 1.3	4.3 ± 0.6	5.5 ± 1.1	8.0 ± 1.9
HV (kgf/mm^2^)	404 ± 125	281 ± 44	480 ± 66	397 ± 163	401 ± 58	508 ± 99	739 ± 176
E (GPa)	112 ± 28	125 ± 14	156 ± 17	136 ± 34	159 ± 18	153 ± 18	157 ± 23
Lc2 (N)	1.66	1.33	1.93	4.66	0.54	0.98	0.85
Lc3 (N)	4.39	4.57	4.22	8.06	3.55	5.01	2.93

## Data Availability

Data sharing not applicable.

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
