# Peer review of "Structural, Mechanical, and Decorative Properties of Sputtered TiN and Ti (N, C) Films for Orthodontic Applications; an In Vitro Study"

_materials, 2021, doi:10.3390/ma14185175_

Round 1
Reviewer 1 Report
COMPLIMENT FOR THE ICONOGRAPHY AND THE SEM IMAGE
PLEASE MODERATE ENGLISH REQUIRED
REFERENCE ARE NOT WRITTEN IN EDITORIAL STANDARD
SEE NOTE IN THE FILE PDF IN ATTACHMENT, WITH SUGGESTION
I WOULD LIKE TO SEE AGAIN THE PAPER
PLEASE IN THE DISCUSSION ADD THE LIMIT OF THIS INTERESTING RESEARCH : THAT IT IS AN IN VITRO STUDY , NOT IN VIVO
SEE PDF IN ATTACHMENT SUGGESTIONS AND CORRECTION
COMPLIMENTI FOR ICONOGRAPHY AND SEM IMAGE
PLEASE DO MINOR CORRECTION ENGLISH ARE REQUIRED
PLEASE IN THE DISCUSSION REPORT THE LIMIT OF THIS STUDY: IN VITRO AND NOT IN VIVO

Author Response
- COMPLIMENT FOR THE ICONOGRAPHY AND THE SEM IMAGE
We thank the referee for this issue, the scale of both images is similar. This information has been inserted in the present version of the manuscript. “Figure 3. Representative SEM images for the set of studied samples. In particular, the left column shows the cross-section and the right column the top view for the A3, B2, and B4 studies films. The scale bar indicated for the cross-section images are the same for the top-view ones.”
- PLEASE MODERATE ENGLISH REQUIRED
We have been reviewed the entire manuscript improving the English as indicated by the referee.
- REFERENCE ARE NOT WRITTEN IN EDITORIAL STANDARD
We have been changed the references accordingly.
- SEE NOTE IN THE FILE PDF IN ATTACHMENT, WITH SUGGESTION
Suggestion #1
We thank the referee the suggestion. We add the reference, however we do not know the DOI, and according with the formatting rules of the Editorial board, we need to know that.
Suggestion #2
We thank the referee the suggestion, however, we do not believe this reference will add new information in the sentence. Again, we do not know the DOI of the work suggested.
- PLEASE IN THE DISCUSSION ADD THE LIMIT OF THIS INTERESTING RESEARCH :
THAT IT IS AN IN VITRO STUDY, NOT IN VIVO
We thank the referee for this comment and, as we discuss in the framework of this paper, “we conduct a systematic study in a simulated oral cavity”, line 95 of the present version.
Reviewer 2 Report
Abstract - minor issues in English language.
Introduction - reasonable, but a lack of clarity due to poor English language.
Please do not use first person 'we', 'us', 'our'
Equations - do not need to be in bold.
Experimental - insufficient detail provided. Should be sufficient for an independent researcher to repeat the study. E.g. XRD exposure time, scan increment. How were the cross sections prepared?
Plots of the variation in hardness would be helpful e.g. as a function of N2 gas flow and/or pellet area.
The English language used throughout this manuscript is unclear and difficult to follow in many places. It is below the standard required for publication - I recommend making use of a text editing service.
The text is at times also imprecise and wordy - please focus on being concise where possible.
Unfilled marker needs to be described in caption of Fig 4b + 5b.
Several of the trends that have been observed cannot be decided from so few data points. In order to make these types of claim, more data needs to be collected, or the claims removed.
Author Response
Responses to the comments of Reviewer #1:

Reviewer 3 Report
Dear Authors,
the topic of the paper may be interesting and appropriate for Materials.
The English language was good and the standard is acceptable for a scientific publication.
the TITLE is a bit long and it should be specified that the study was conducted in vitro.
The AIM is clearly stated and the INTRODUCTION places the study in the right perspective, although a greater compared with the literature would be appreciated.
The MATERIALS AND METHODS section, although quite clearly describe, has some lacks:
- Please in the statistical analysis section, specify what each calculation was used for.
- In the statistical analysis, the Method Error is not calculated. no information about the period of recruitment is specified.
The RESULTS and DISCUSSION address the original objective of the study, are relates to the findings, are well elaborated and exhaustive but should be slightly reduced
The CONCLUSIONS briefly exposed the main findings of the study.
Please consider adding this reference to yours:
https://pubmed.ncbi.nlm.nih.gov/32106415/
Author Response
Answer to the Reviewer 2.

Reviewer 4 Report
The paper is rather good-quality and well-organized. Nevertheless there are some remarks, which are listed below:
- For the gas flow rate please use rather SLPM as an unit.
- For Ti(N,C) films the difference between 2.4 and 2.5 sccm is to small in my opinion - please explain it.
- XRD analysis - what was the range, scan step and time for step during XRD?
- Scratch test - what was the value of the maximum load? What was the scratch length? How many scratches for one sample have been carried out?
- Hardness - how many indents have been carried out?
- Sliding test - why Authors used steel ball against ceramic one (e.g. WC)? Additional, what was the sliding radius? Moreover, the distance is too short, friction conditions did not have time to stabilize.
- On the top view (Fig. 3) there is no scale bar (or maybe it is the same as for cross-sections, but I did not find this information).
- Elastic modulus value (112 GPa) is out f range, on which the Authors reffer to (line 254) - please explain.
- Hardness value equal to 8 GPa - is it an average value? Please add standard deviation value.
- Table 2 - there is no image after scratch test - please add it and comment the value of critical loads. Also add standard deviation values.
- Wear rate values - please uniform the accuracy.
- And last but not least - it is definitely not similar thickness: 400 nm and 1200 nm it is 3 times higher (or lower). So the comparison coatings with such big difference in the thickenss is problematic form a scientific and engineering point of view.
Author Response
Answer to the Reviewer 3.

Round 2
Reviewer 1 Report
Lucchese, A.; Carinci, F.; Brunelli, G.; Monguzzi, R. Everstick® and Ribbond® fiber reinforced composites: Scanning Electron Microscope (SEM) comparative analysis. Eur J Inflamm. 2011, 73-9.suggest to add in Line 32, regarding manufacturing technology. I think is appropriate to add this references , line 32 , manufacturing snd technology